# Fool's-gold science

*The ethical and scientific perils of testing most vaccines using placebo-controlled randomized trials*

Arthur Caplan [1✉], Felicia L Pasadyn [2] & Nathaniel Mamo[2]

See also: B Pulverer

The highest health official in the USA, Secretary of Health and Human Services (HHS) Robert F. Kennedy Jr., recently said, "[t]he only vaccine that was ever safety tested in a clinical trial against a placebo was the COVID vaccine. None of the others have any kind of long-term testing or even any testing." Since he has been sworn in, HHS has adopted his ideas as official policy, repeating his statements almost verbatim, "except for the COVID vaccine, none of the vaccines on the CDC's childhood recommended schedule was tested against an inert placebo, meaning we know very little about the actual risk profiles of these products."

HHS further announced that, "All new vaccines will undergo safety testing in placebo-controlled trials prior to licensure–a radical departure from past practices." Although it is unclear how it will be implemented, this policy promises to upend decades of scientific work, and its announcement is already undermining trust in vaccines and science.

But the proposed imposition of placebo-controlled randomized clinical trials (RCTs) as the gold standard is nothing but pyrite —"fool's gold". Kennedy Jr.'s views and HHS policy ignore both reality and history. All vaccines undergo a lengthy process of careful testing and independent regulatory oversight in the USA and other nations. Many versions have actually been tested in placebo-controlled RCTs. The HHS policy would be harmful in that it would impose huge additional costs and delay vaccine improvement and access. Moreover, it would violate internationally accepted ethical principles.

> *... the proposed imposition of placebo-controlled randomized clinical trials (RCTs) as the gold standard is nothing but pyrite —"fool's gold".*

There is an important distinction between testing variant formulations or improved versions of already approved vaccines—such as a new strain-targeting HPV vaccine—and testing entirely novel vaccines for diseases with no previous immunization. In the former case, active-controlled non-inferiority trials are both scientifically valid and ethically appropriate to avoid unnecessary harm to participants. In contrast, novel vaccines such as the first HPV vaccine in the early 2000s required placebo-controlled trials because no preventative option existed at the time. By ignoring this distinction, current HHS policy threatens to delay critical incremental improvements in vaccine technology, disrupt supply chains, and ultimately reduce timely access to life-saving immunizations. Requiring placebos for improved vaccine variants would needlessly expose research participants to deadly diseases for which effective vaccines are available. Placebo testing is therefore not appropriate for already approved vaccines and should only be done when no other options exist and a fatal epidemic is looming against which only a novel vaccine would be available.

> *... current HHS policy threatens to delay critical incremental improvements in vaccine technology, disrupt supply chains, and ultimately reduce timely access to life-saving immunizations.*

## The actual history of RCTs in vaccine testing

A favorite target of anti-vaccine activists—astonishing, given the near-eradication of a devastating disease through decades of global vaccination efforts—is the polio vaccine. Close and current advisors to Kennedy Jr. have, for years, petitioned the US Federal Drug Administration (FDA) to withdraw its approval on grounds that it was inadequately tested. Similar allegations about a lack of RCTs impugning previous safety assessments have been brought against other well-established vaccines against human papillomavirus (HPV), hepatitis B, and mumps (Schiller et al, 2012; Vogel, 2025).

The facts about testing the first polio vaccine are indisputable. On April 12, 1955, the results of the largest public health experiment in history were announced. Beginning in 1954, Jonas Salk tested his inactivated polio vaccine (IPV) on himself and then on nearly two million children in a nationwide randomized, double-blinded, placebo-controlled trial (Oshinsky, 2005). After meticulous review from public health

[1]Division of Medical Ethics, NYUGSOM, New York, NY 10016, USA. [2]NYU Grossman School of Medicine, New York, NY 10016, USA. ✉E-mail: Arthur.caplan@nyumc.org
https://doi.org/10.1038/s44319-025-00530-5 | Published online: 31 July 2025

authorities, the vaccine was determined to be safe and effective as it conferred 80–90% protection against polio, a disease which just a year before had no method of prevention whatsoever (Oshinsky, 2005).

Even without placebo-controlled RCTs, other polio vaccines have been rightfully trusted to be safe and effective. Huge observational field trials of Sabin's oral polio vaccine (OPV), were carried out in the USSR first on 20,000 children in 1958, and then on 10 million Soviet children in 1959, as well as upon 110,000 Czechoslovakian children in the same year. The data from these trials, comparing vaccinated and non-vaccinated populations, proved beyond doubt that this vaccine was safe and effective, just like IPV (Horstmann, 1991). Many subsequent studies using blinded but not placebo-controlled RCTs were done in Cuba, Oman, the Philippines, and India to refine dosing and scheduling along with various comparator trials (Murph et al, 1987; Plotkin et al, 2018).

The near-universal uptake of these highly effective vaccines underscores the strong confidence of vaccine trials in the scientific community and the public. It is simply a politically fueled canard to deny the key role of placebo-controlled RCTs, as well as the legitimacy of other trial types, in the development of polio and other vaccines (Chisti, 2025).

........................................................

*It is simply a politically fueled canard to deny the key role of placebo-controlled RCTs, as well as the legitimacy of other trial types, in the development of polio and other vaccines.*

........................................................

## Is a placebo-controlled RCT the only way to generate valid evidence about vaccine safety and efficacy?

The suggestion by Kennedy Jr. and some of his supporters that the only valid method for producing credible evidence is a placebo-controlled RCT is false (Vogel, 2025). There are a variety of well-established methods that researchers in medicine regularly use to reliably test a scientific question. They form the basis of what scientific experts consider to be the rightful "gold standard" of science (Kratsios, 2025).

One of the most fundamental scientific techniques is systematic observation, often using calibrated instruments. Fields ranging from anatomy to clinical diagnostics rely heavily on observational data to identify disease states or symptom patterns. These observations—when reproduced across sufficiently large cohorts to achieve statistically robust conclusions—can yield reliable data without the need for randomization or placebo controls.

Similarly, by watching groups of individuals or cohorts over time, historical or longitudinal studies can reveal associations or patterns that tie diseases—such as lung cancer or osteoarthritis—to particular causes: smoking, vigorous repetitive exercise, or obesity. There are many such studies showing the reduction and elimination of polio all over the world that are acknowledged as sound and reliable.

Research using interviews, observations and monitoring can provide in-depth, trustworthy understanding of many disease phenomena. Surveys can gather data from a large number of participants, providing insights into symptoms, disease remission and adverse events without a placebo being involved.

## Why placebo controls are very rarely ethical in vaccine trials

An RCT is ethically justifiable only if there is a state of "equipoise," meaning genuine uncertainty or lack of consensus within the medical community about which intervention is the most efficacious. Otherwise, ethically, the control group in an RCT must receive the highest "standard of care" (WMA, 2025). This ensures that participants in the control group are not disadvantaged or exploited by being denied known effective or partially effective treatments.

The exposure of the unethical Tuskegee syphilis study, where a placebo arm was maintained to observe the natural progression of untreated syphilis in a group of impoverished African-American men despite the availability of curative treatment, led to a profound reckoning in biomedical research ethics. American law, international ethical frameworks, and binding professional codes recognized the inviolable duty of researchers to offer the best available treatment options in studies involving illness or risk of harm (Wolinsky, 1997; CIOMS 2017; WMA Declaration of Helsinki, 2025).

Calls to conduct retrospective placebo-controlled trials on well-established interventions such as the polio vaccine—or prospective trials that withhold proven vaccines in order to test minor improvements—are therefore scientifically unnecessary, transgressing universal ethical principles. When the safety and efficacy of a vaccine have already been rigorously demonstrated, withholding them in the name of a placebo comparison violates both modern ethical standards and the basic tenets of human subject protection.

In addition, risk reduction is mandatory for all studies, even if it creates more challenges to demonstrate an agent as safe and reliable. For polio, this means always studying modified vaccines against existing approved vaccines and encouraging all subjects to maintain good hygiene and avoid travel to areas afflicted with outbreaks to reduce the risk of infection.

In an unexpected public health crisis, RCTs may not be required prior to vaccine approval. When facing a rapidly spreading and fatal disease, and when patients are at risk of dying before an RCT could reasonably be conducted, emergency use authorization or compassionate use may be ethically justified. The ethical justification for approving a therapy, including a vaccine, without the usual body of supporting evidence hinges on several key conditions: a disease causing substantial morbidity and mortality; no known preventive or therapeutic interventions; informed consent from vaccine recipients acknowledging the vaccine is untested; and a robust plan to monitor outcomes through observational studies.

This framework was applied during the COVID-19 pandemic, when mRNA vaccines from Pfizer/BioNTech and Moderna—based on a novel technology platform—were deployed under emergency use authorization. In contrast to traditional vaccines that rely on decades-old and well-established methods using attenuated pathogens or recombinant proteins, mRNA vaccines represented a major technological advancement. Their rollout highlighted both the ethical complexity and scientific urgency of responding to a global health emergency, while still preserving rigorous post-authorization monitoring and follow-up studies to ensure safety and efficacy. These exceptional circumstances bear no relevance to Kennedy Jr.'s calls for placebo-controlled RCTs for the polio or other long-established, safe and effective vaccines.

## Conclusion

Calls or policies requiring the use of placebos to test vaccines are ethically flawed and unsound in their insistence that this is the only method for generating credible or so-called 'gold standard' science. The American Academy of Pediatrics has already stated that it will not follow these demands, rather issuing their own evidence-based guidelines. Essentially, Secretary Kennedy Jr. and HHS are asking the American public to start from scratch again, testing vaccines that have already been proven safe and effective. Far from upholding scientific rigor, such demands distort the purpose of trials and jeopardize public health (Gostin and Reiss, 2025). They hasten the erosion of herd immunity, already under strain for diseases such as measles due to declining vaccination rates. Rebuilding public confidence and restoring coverage will take years, if not decades. No sponsor, investigator, research ethics committee, manufacturer, or regulator ought to participate in trials that disregard existing evidence and indefensibly expose populations to preventable harm under the false guise of scientific purity. Demanding such trials for proven interventions is not only scientifically unsound but also ethically indefensible.

.......................................................

*Essentially, Secretary Kennedy Jr. and HHS are asking the American public to start from scratch again, testing vaccines that have already been proven safe and effective.*

.......................................................

## Peer review information

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

## Disclosure and competing interests statement

The authors declare no competing interests.

