## [Peer Review File · EMBO Reports]

The Dubious Ethics of Testing Most Vaccines Using Placebo-Controlled Randomized Trials

Arthur Caplan, Nathaniel Mamo, and felicia Pasadyn

Corresponding author: Arthur Caplan (Arthur.Caplan@nyumc.org)

Review Timeline:

Submission Date:

6th Jul 25

Accepted:

10th Jul 25

Editor: Holger Breithaupt

Transaction Report:

This article was editorially reviewed.

Prof. Arthur Caplan
NYU Langone Medical Center
Division of Medical Ethics
227 East 30th Street
New York, NY 10016
United States

Dear Prof. Caplan,

I am pleased to accept your Science & Society article for publication in the next available issue of EMBO reports.

Your manuscript will be processed for publication by EMBO Press. It will be copy edited and you will receive page proofs prior to publication. Please note that you will be contacted by Springer Nature Author Services to complete licensing and information.

It has been a pleasure to work with you on this article. Thank you for contributing to EMBO reports.

Sincerely,

Holger Breithaupt, PhD
Senior Editor, Science & Society
EMBO reports

Prof. Arthur Caplan
NYU Langone Medical Center
Division of Medical Ethics
227 East 30th Street
New York, NY 10016
United States

Dear Prof. Caplan,

I am pleased to accept your Science & Society article for publication in the next available issue of EMBO reports.

Your manuscript will be processed for publication by EMBO Press. It will be copy edited and you will receive page proofs prior to publication. Please note that you will be contacted by Springer Nature Author Services to complete licensing and information.

It has been a pleasure to work with you on this article. Thank you for contributing to EMBO reports.

Sincerely,

Holger Breithaupt, PhD
Senior Editor, Science & Society
EMBO reports